# Could Evening Dietary Protein Intake Play a Role in Nocturnal Polyuria?

**DOI:** 10.3390/jcm9082532

**Published:** 2020-08-05

**Authors:** Upeksha S. Alwis, Joris Delanghe, Lien Dossche, Johan Vande Walle, John Van Camp, Thomas F. Monaghan, Saskia Roggeman, Karel Everaert

**Affiliations:** 1Department of Human Structure and Repair, Ghent University, 9000 Ghent, Belgium; karel.everaert@ugent.be; 2Department of Diagnostic Sciences, Ghent University, 9000 Ghent, Belgium; Joris.Delanghe@ugent.be; 3Department of Internal Medicine and Pediatrics, Ghent University, 9000 Ghent, Belgium; lien.dossche@ugent.be (L.D.); Johan.VandeWalle@uzgent.be (J.V.W.); 4Department of Pediatric Nephrology, Ghent University, 9000 Ghent, Belgium; 5Department of Food Technology, Safety and Health, Ghent University, 9000 Ghent, Belgium; John.VanCamp@ugent.be; 6Department of Urology, SUNY Downstate Health Sciences University, Brooklyn, NY 11203, USA; thomas.monaghan@downstate.edu; 7Research and Policy Department, Psychiatric Center Sint-Jan-Baptist, 9060 Zelzate, Belgium; saskiaroggeman@gmail.com

**Keywords:** urea excretion, diuresis, nocturnal polyuria, sodium excretion, dietary protein

## Abstract

Urea is the most abundant and the largest contributing factor for urine osmolality. Urinary urea excretion is highly interrelated with dietary protein intake. Accordingly, an increase of urinary urea excretion due to high protein diet may lead to urea-induced osmotic diuresis. This study aims to explore the association between nocturnal polyuria (NP) and urea. This is a post hoc analysis of a prospective observational study of subjects who completed a renal function profile between October 2011 and February 2015 (n = 170). Each subject underwent a 24 h urine collection, which included 8 urine samples collected at 3 h intervals. Urine volume, osmolality, creatinine, urea and sodium were determined. Urinary urea excretion was used to estimate dietary protein intake. Compared to the control group, subjects with NP exhibited significantly higher nighttime urea and sodium excretion. Estimated evening dietary protein intake was correspondingly significantly higher amongst the NP subgroup. Nighttime diuresis rate was positively associated with age and nighttime free water clearance, creatinine clearance, sodium excretion, and urea excretion in NP subjects. Therefore, increased nocturnal urinary urea excretion may reflect an additional important mediator of nocturia owing to excess nocturnal urine production.

## 1. Introduction

Nocturia, or the need to wake up to urinate during the main sleep period, is a highly prevalent and bothersome lower urinary tract symptom [1]. Nocturia-mediated sleep disruption impairs quality of life and overall health, and nocturnal voiding has been associated with a higher incidence of cardiovascular disease, falls and fractures in older adults, and mortality [2]. Nocturnal polyuria (NP), or excess urine production during the main sleep period [1], is thought to be the most common cause of nocturia in adults of all ages [3,4].

Recent research has shown that NP is a heterogeneous condition that may be driven by excess free water and/or osmotic diuresis [5], and impaired circadian rhythm in renal handling of free water and sodium has been observed in both adults [6] and children [7] with NP. Thus current pharmacologic interventions for management of NP involves concomitant medication based on the NP phenotype; such as antidiuretic therapy [6,8]. However, one-third of patients shows resistant to the current therapeutic strategies [7,9], which might suggest a possible effect involved in lifestyle; such as diet. Therefore, lifestyle interventions targeting both nocturnal free water and sodium diuresis have garnered considerable attention in the management of nocturia owing to NP [9,10,11,12]. However, relatively less attention has been afforded to urea, which, alongside sodium, is the primary constituent of urine osmolality on molar basis [13].

Urea is a small organic molecule produced in the liver, which plays an important role in body nitrogen balance [14]. It is the primary nitrogenous waste product in the catabolism of dietary proteins [13,14]. Urinary urea excretion is known to be highly interrelated with dietary protein intake [13,14,15,16]. While a small amount of urea can be excreted through sweat and used by the gut microbiota [17], majority of the urea produced in the liver is filtered by the kidneys and eliminated in urine [18]. Urea can freely pass from the blood into the glomerular filtrate, and thus urea concentration in the proximal tube of the nephron is determined by the glomerular filtration rate (GFR). Along the tubule of the nephron, urea is both secreted and reabsorbed, mediated by the urea transport proteins, which consequently results in approximate 50% of filtered urea to excrete in urine [15]. The degree of urea reabsorption may affected by urine flow rate, GFR, and dietary protein intake [19].

A typical western diet with approximately 80 g of protein/day accounts for a daily urinary excretion of 950 of total milliosmoles, 400 mmol of urea, 140 mmol of sodium, and other solutes in healthy adults [13]. Accordingly, daily urea excretion becomes far greater than daily sodium excretion, which accounts for almost half of the urine osmolality. As such, excretion of a large quantity of urea, following a significant dietary protein intake in the evenings, may reflect an additional important mediator in the pathogenesis of NP. However, available data regarding the effect of urea derived from dietary protein on urine production is limited and remains controversial [20,21].

Thus, the present study aimed to explore the role of urea in pathogenesis of NP.

## 2. Materials and Methods

### 2.1. Study Design and Population

The present study is a post hoc analysis of data obtained from a prospective observational protocol designed to explore circadian changes in renal function conducted between October 2011 and February 2015. Participants were recruited from the local hospital, university, and community via posters, flyers, lectures, and consultations. Individuals with neurological disorders, signs of congestive heart failure, diuretic use, or renal failure were excluded. The study protocol was approved by the Ghent University Hospital Ethics Committee (EC2011/565) and conducted in accordance with the Declaration of Helsinki. Written informed consent was obtained from all subjects.

### 2.2. Renal Function Profiling

Each subject underwent a 24 h urine collection, which included 8 samples (U1–U8) collected at 3 h intervals [5]. The first urine sample, U1, was collected between 8–11 a.m., U2 between 11 a.m.–2 p.m., U3 between 2–5 p.m., U4 between 5–8 p.m., U5 between 8–11 p.m., U6 between 11 p.m.–2 a.m., U7 between 2–5 a.m., and U8 between 5–8 a.m. U1–U5 were classified as daytime samples and U6–U8 were classified as nighttime samples [5].

Diuresis rate, osmolality, creatinine, urea, and sodium concentrations were determined. The 24 h urinary collection was confirmed by using the reference range for 24 h urinary creatinine excretion mg/kg body weight for males (20 to 25 mg/kg) and females (15 to 20 mg/kg) [22]. Osmolality was measured using freezing-point depression (OSMO station OM–6060, Arkray) [23]. Creatinine was measured using a compensated rate-blanked alkaline picrate method [24]. Urea was measured using an enzymatic assay (urease/glutamate dehydrogenase method) [25]. Sodium was measured using indirect potentiometry (Cobas 8000 modular analyzer, Roche) [26].

A blood sample was taken from each participant and serum osmolality, creatinine, urea and sodium were utilized to derive renal solute clearance (urine_osmolality_ × urine flow/plasma_osmolality_), free water clearance (urine flow–solute clearance) and creatinine clearance (urine_creatinine_ × urine flow/plasma_creatinine_) (5). Absolute urinary excretion of sodium (urine_sodium_ × urine volume) and urea (urine_urea_ × urine volume) were calculated. The fractional excretion of solutes (urea and sodium) was calculated as {(urine_solute_ × plasma_creatinine_)/(plasma_solute_ × urine_creatinine_) × 100}.

### 2.3. Protein Intake Estimation

Absolute urinary urea excretion was employed to estimate dietary protein intake ([urinary urea nitrogen excretion (UUN; g/day)] + 0.031 × [body weight (kg)] × 6.25) [27].

### 2.4. Statistical Analysis

Patients were compared by NP status using a NP cutoff of nocturnal polyuria index (nocturnal urine volume/24 h total urine volume) > 0.33 in accordance with current International Continence Society terminology (1). Clinical and biochemical parameters were compared using the chi-square test and Mann-Whitney U test for categorical and continuous variables, respectively. All continuous measures are reported as median (interquartile range).

Multiple linear regression analysis was used to evaluate the relationship between nighttime diuresis rate and age, body mass index, sex, free water clearance, creatinine clearance, and sodium and urea excretion. Pearson correlation coefficients were used to assess the relationship between age and plasma urea concentration and urinary urea excretion.

All data were analyzed using IBM SPSS statistics for Windows (version 25.0, IBM Corp, Armonk, NY, USA). A *p*-value < 0.05 was deemed statistically significant.

## 3. Results

A total of 170 adults were eligible for analysis (62% female, median age 66 (51–72) years), of which 118 (69%) met the cutoff for NP (Table 1). Twenty-four hour urinary creatinine output confirmed no errors in 24 h urine collection.

### 3.1. Renal Function Profiles and Estimated Protein Intake

Compared to the controls, subjects with NP exhibited lower urine osmolality and higher sodium excretion during the nighttime. Conversely, during the daytime, NP subjects demonstrated a higher urine osmolality and lower sodium excretion. No significant differences in 24 h urine osmolality or sodium excretion were observed between groups.

Analogous to the renal function profile data, nighttime urea excretion and estimated evening protein intake were both higher in the NP group, among whom daytime urea excretion and estimated daytime protein intake were also lower than in controls. Likewise, no significant differences in 24 h urea excretion or estimated 24 h protein intake were observed between groups.

Figure 1 shows the absolute urinary excretion of urea and sodium for the subjects with and without NP over 24 h. As shown in the figure, 24 h urinary excretion of urea was greater than sodium excretion in both groups. Absolute urinary urea excretion was inversely correlated with age, in the NP group (r = −0.300, *p* = 0.001) and in the control group (r = −0.492, *p* < 0.001). No significant correlation was observed in plasma urea concentration and age in the NP group, but plasma urea concentration and age was positively correlated in the control group (r = 0.405, *p* = 0.003).

In multiple regression analysis, nighttime diuresis rate was positively associated with age and nighttime free water clearance, creatinine clearance, sodium excretion, and urea excretion in the NP group (Table 2). In the control group, nighttime diuresis rate was also positively associated with nighttime free water clearance, creatinine clearance, sodium excretion, and urea excretion.

### 3.2. Circadian Rhythms of Water and Osmotic Diuresis

Figure 2 shows the circadian rhythm of diuresis rate (a), free water clearance (b) osmolality (c) and solute clearance (d) for the subjects with and without NP over 24 h. In the early night collection (U6), subjects with NP exhibited significantly higher diuresis rate, free water clearance, and solute clearance, whereas urinary osmolality was significantly lower compared to the controls. In the controls, diuresis rate and solute clearance were lower and urine osmolality was higher during the nighttime, compared to the daytime. This kind of circadian rhythm was not observed in the NP group.

As shown in Figure 3, (a) and (b) subjects with NP exhibited a loss of circadian rhythms in both absolute and fractional sodium excretions. No significant difference was observed in nighttime collection of urinary sodium concentrations between the two groups (Figure 3c). Urine sodium–to–osmolality ratio remained higher throughout the nighttime (Figure 3c). In contrast, in the control group, sodium excretions and urine sodium–to–osmolality ratio dropped largely during the nighttime, showing a clear difference between day and night.

Compared to the controls, NP subjects also exhibited a blunted circadian rhythm in urea excretion (Figure 4a,b), wherein both absolute and fractional urea excretions during the nighttime remained higher. Urine urea-to-osmolality ratio lacked circadian rhythm in the NP group and showed lower urea-to-osmolality ratio during the night, compared to the control group (Figure 4c).

## 4. Discussion

NP is a heterogeneous condition that exists of three subtypes based on its etiology; free water-predominant diuresis, solute-predominant diuresis, or mixed diuresis [5]. Loss of circadian rhythm in renal handling of free water in water diuresis and sodium in osmotic diuresis has been reported in earlier studies with NP patients [6,7].

In the present study, we observed that daytime urea excretion was significantly lower in the NP group, compared to the control group. In contrast, the nighttime urea excretion was significantly higher in the NP group. No significant difference was observed in 24 h osmotic or urea-load. Similarly, during the early night urine collection (U6: 11 p.m. to 2 a.m.), fractional urea excretion significantly increased in NP subjects. Furthermore, the nocturnal diuresis rate was positively associated with the nighttime urea excretion. Taken together, this phenomenon of NP and the loss of circadian rhythm in urea excretion (see Figure 4) may reflect an additional important mediator in nocturia owing to excess nocturnal urine production.

In the previous studies, consumption of a large protein-rich diet has been associated with increased urinary urea excretion [14,21], increased blood urea levels [17], a period of glomerular hyper-filtration [20,21], and increased urinary sodium excretion [20,21]. Unlike other macronutrients, protein digestion increases amino-acid derived metabolites as urea and other metabolites (phosphates, sulfates, etc.), which are mainly excreted by the kidney [16,17]. Additionally, a high protein diet is often a source of excess sodium, potassium, phosphates, etc. [15]. Therefore, dietary protein intake is considered as the main determinant of potential renal solute load [17]. In Belgium, the main sources of dietary protein come from meat and meat products (34.6%), grain and cereal products (21.4%), and milk and substitute products (19%) [28]. Likewise, meat and meat products (26%), grain and cereal products (25%), and milk and substitute products (14%) are also the major sources of dietary sodium intake in Belgium [29]. Therefore, to illustrate the potential effect on urine osmolality by consuming a high-protein diet, we calculated the approximate osmotic load (mOsm) due to urea, based on protein content, and sodium content of common protein-rich foods (Table 3). As illustrated by Table 3, excretion of excess amount of urea and sodium, followed by consuming a protein-rich diet, especially from processed meat, dairy, or cereal products, might induce excess diuresis due to osmotic load.

In the present study, we observed that evening dietary protein intake, as reflected in the urinary urea-based estimate of dietary protein intake, was correspondingly significantly higher amongst the NP group, as would be consistent with a large, protein-rich evening meal. However, this is based on the urea-based calculation of protein intake, where the 2 parameters are in the formula as such directly correlated. Therefore, at the absence of food intake data in our study, we cannot conclude that the higher nocturnal urea excretion in the NP group is due to a large protein-rich evening meal. However, urine urea nitrogen based estimation of dietary protein intake has been useful in the absence of dietary intake history [30,31,32].

In the present study, the median age of the NP group was significantly high and the nocturnal diuresis rate was positively associated with the age in the NP group. Therefore, evening high-protein intake and urea excretion might be more important in the elderly patients with NP, owing to few reasons. First, the maximal renal concentrating capacity diminishes with aging [33] and age-related decrease in fractional urea excretion in subjects with normal renal function has been reported earlier [18,34]. In the present study, age-related decrease in urinary urea excretion was also observed in both groups. Secondly, incorporation of protein in muscle tissue may lower with advancing age, thus protein metabolism may be well different for younger versus older people [17].

In addition to these results, a greater proportion of males was consisted in the NP group, which might be related to the previously reported sex differences in urine concentration activity and/or osmotic load due to different food intake [35].

Nevertheless, NP is a heterogeneous clinical condition which is more complicated than food intake. As shown in Figure 2, Figure 3 and Figure 4, loss of renal circadian rhythms in free water and solute handling plays a role. Yet, the possible effect of food (protein/salt intake) and meal timing in the pathogenesis of NP is important, especially for the patients showing resistance to the antidiuretic therapy [7,9]. A growing amount of evidence from the emerging field of *Chrononutrition* have suggested the effect of meal time on metabolism and the endogenous circadian rhythms in both human and rodent studies [36,37]. Therefore, future studies are needed to address the dietary effect on disturbed circadian rhythm in the pathogenesis of NP.

The present study was an post hoc analysis and limited to a relatively small study population. No food intake data was available for the study subjects; thus, the estimated protein intake was calculated based on urinary urea excretion. Likewise, we assumed that the total urea excretion was solely derived from the dietary protein intake and urea was only excreted via the kidney and no other routes. Apart from these study limitations, to our knowledge, this study showed a novel additional effect of urea on NP. The study design to evaluate eight urine samples collected over 24 h helped identify the trends in renal functions over time using repeated samples.

Urine volume primarily depends on the maximal concentration ability of the kidney, along with the macronutrients and salt content of the diet, and the amount of metabolites to be excreted [38]. Therefore, future research on the association between NP and diet is warranted. On the other hand, the relationship between urea excretion and vasopressin needs to be explored. Impaired circadian rhythm of vasopressin in NP has been correlated with loss of maximal concentration ability of the kidney [7]. During water diuresis, a rapid increase in urea excretion occurs, which is explained by decreased urea reabsorption in the papillary collection duct and loss of urea from the renal medulla [19].Thus, the finding of increased urea excretion during the nighttime in the NP group could also be explained by a decrease in vasopressin levels at night in the NP group.

## 5. Conclusions

In the present study, higher nighttime urea excretion was observed in the NP group. In the NP group, the nighttime diuresis rate was positively associated with age and nighttime free water clearance, urea and sodium excretion, and creatinine clearance. Taken together, the results of this study suggest that, besides free water and sodium, urea also plays an additional important role in nocturnal urine production and the occurrence of nocturia. A reduction of evening protein consumption may be an effective lifestyle intervention in the management of nocturia owing to NP.

## Figures and Tables

**Figure 1 jcm-09-02532-f001:**
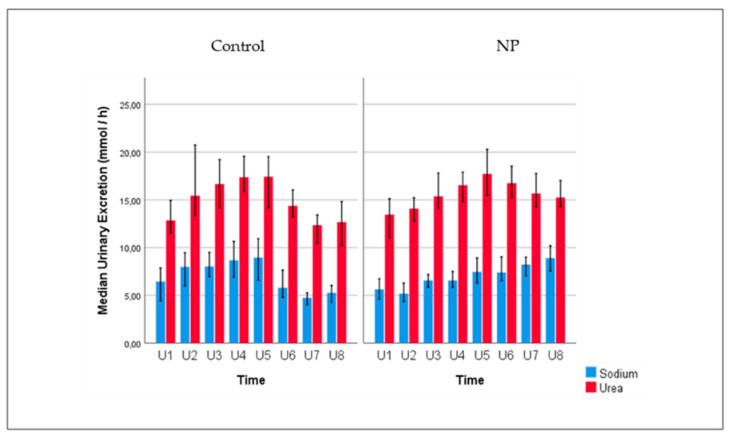
Urinary excretion of urea and sodium (mmol/hour) for subjects with and without NP (control) according to eight urine samples collected over 24 h. Daytime urine samples were collected between 8–11 a.m. (U1), 11 a.m.–2 p.m. (U2), 2–5 p.m. (U3), 5–8 p.m. (U4), and 8–11 p.m. (U5). Nighttime urine samples were taken at 11 p.m.–2 a.m. (U6), 2–5 a.m. (U7), and 5–8 a.m. (U8). Error bars: 95% CI. **Note:** As shown in the figure, on molar basis, urea is the most abundant urinary solute.

**Figure 2 jcm-09-02532-f002:**
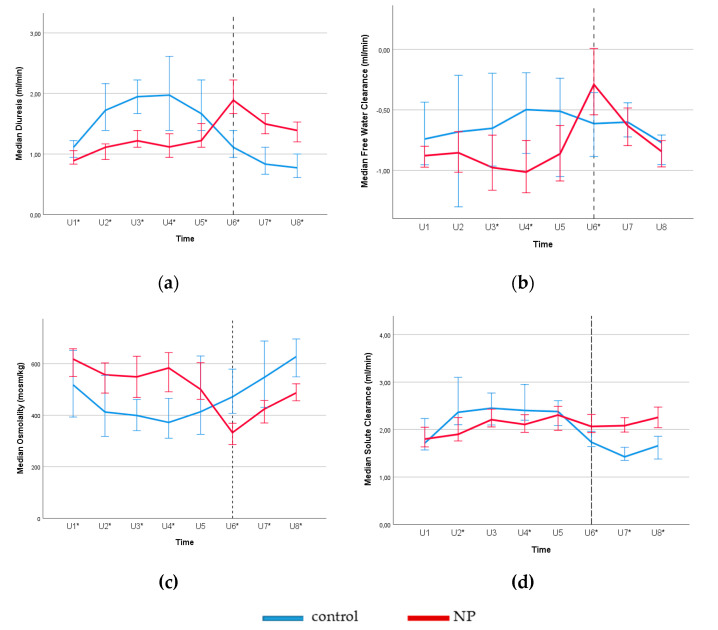
Circadian rhythms for diuresis rate (**a**), free water clearance (**b**), osmolality (**c**) and solute clearance (**d**) for subjects with and without NP (control) according to eight urine samples collected over 24 h. Reference line separates the day and night time. Daytime urine samples were collected between 8–11 a.m. (U1), 11 a.m.–2 p.m. (U2), 2–5 p.m. (U3), 5–8 p.m. (U4), and 8–11 p.m. (U5). Nighttime urine samples were taken at 11 p.m.–2 a.m. (U6), 2–5 a.m. (U7), and 5–8 a.m. (U8). * *p* < 0.05 for NP versus the control group (Mann-Whitney U test). Error bars: 95% CI. **Note:** In the early night, collection subjects with NP shows higher diuresis rate, free water clearance, solute clearance and lower urine osmolality, which shows the loss of circadian rhythm in renal handling of free water in water diuresis.

**Figure 3 jcm-09-02532-f003:**
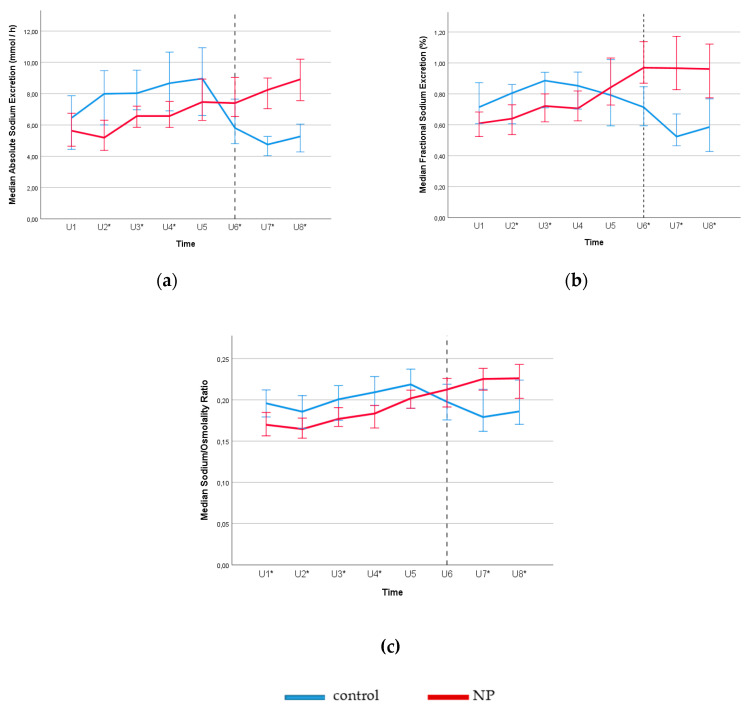
Circadian rhythms for absolute sodium excretion (**a**), fractional sodium excretion (**b**) and sodium/osmolality ratio (**c**) for subjects with and without NP (control) according to eight urine samples collected over 24 h. Reference line separates the day and night time. Daytime urine samples were collected between 8–11 a.m. (U1), 11 a.m.–2 p.m. (U2), 2–5 p.m. (U3), 5–8 p.m. (U4), and 8–11 p.m. (U5). Nighttime urine samples were taken at 11 p.m.–2 a.m. (U6), 2–5 a.m. (U7), and 5–8 a.m. (U8). * *p* < 0.05 for NP versus the control group (Mann–Whitney U test). Error bars: 95% CI. **Note:** Subjects with NP shows increased sodium excretion throughout the night, which shows the loss of circadian rhythm in renal handling of sodium in osmotic diuresis.

**Figure 4 jcm-09-02532-f004:**
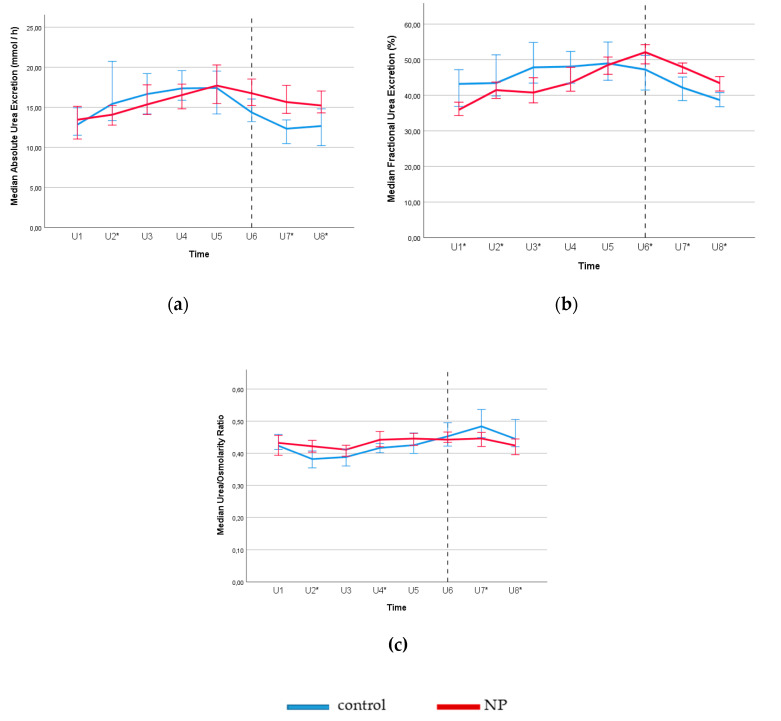
Circadian rhythms for absolute urea excretion (**a**), fractional urea excretion (**b**) and urea/osmolality ratio (**c**) for subjects with and without NP (control) according to eight urine samples collected over 24 h. Reference line separates the day and night time. Daytime urine samples were collected between 8–11 a.m. (U1), 11 a.m.–2 p.m. (U2), 2–5 p.m. (U3), 5–8 p.m. (U4), and 8–11 p.m. (U5). Nighttime urine samples were taken at 11 p.m.–2 a.m. (U6), 2–5 a.m. (U7), and 5–8 a.m. (U8). * *p* < 0.05 for NP versus the control group (Mann–Whitney U test). Error bars: 95% CI. **Note:** Subjects with NP shows higher urea excretion during the night time compared to the controls, which shows a blunted circadian rhythm in renal handling of urea in osmotic diuresis.

**Table 1 jcm-09-02532-t001:** Study population characteristics.

	Control (n = 52)	NP (n = 118)	*p*–Value
Age (years)	51 (37–66)	67 (58–73)	<0.001 *
Sex (F // M)	40 // 12	65 // 53	0.007 *
BMI (kg/m^2^)	24 (22–26)	25 (22–28)	0.172
NUP (ml/h)	56 (44–84)	100 (71–126)	<0.001 *
Daytime osmolality (mOsm/kg)	435(318–577)	537 (442–684)	0.001 *
Nighttime osmolality (mOsm/kg)	547 (402–680)	411 (337–534)	0.001 *
24-h osmolality (mOsm/kg)	470 (382–622)	505 (412–628)	0.404
Daytime sodium excretion (mmol)	144 (101–175)	108 (69–137)	<0.001 *
Nighttime sodium excretion (mmol)	53 (36–73)	80 (61–102)	<0.001 *
24-h sodium excretion (mmol)	199 (147–259)	188 (140–247)	0.361
Daytime urea excretion (mmol)	306 (233–371)	260 (198–324)	0.009 *
Nighttime urea excretion (mmol)	132 (106–152)	156 (129–197)	<0.001 *
24-h urea excretion (mmol)	442 (372–558)	413 (358–538)	0.351
Daytime protein intake (g/kg body weight)	0.9 (0.8–1.1)	0.8 (0.6–0.9)	<0.001 *
Evening protein intake (g/kg body weight)	0.5 (0.4–0.6)	0.6 (0.5–0.6)	0.005 *
24-h protein intake (g/kg body weight)	1.3 (1.1–1.6)	1.2 (1.0–1.4)	0.063

NP, nocturnal polyuria; BMI, body mass index; NUP, nocturnal urine production. * *p* < 0.05 for χ^2^ test/Mann–Whitney U test.

**Table 2 jcm-09-02532-t002:** Multiple regression analysis for risk factors associated with nighttime diuresis rate.

	Control		NP	
	β	*p*−Value	β	*p*−Value
Age (years)	0.120	0.070	0.117	0.001 *
Gender	−0.039	0.491	−0.015	0.629
BMI (kg/m^2^)	0.028	0.596	0.021	0.466
Nighttime free water clearance (mL/min)	1.1016	<0.001 *	0.971	<0.001 *
Nighttime sodium excretion (mmol)	0.595	<0.001 *	0.509	<0.001 *
Nighttime urea excretion (mmol)	0.461	<0.001 *	0.497	<0.001 *
Nighttime creatinine clearance (mL/min)	0.217	0.006 *	0.213	<0.001 *

NP, nocturnal polyuria; BMI, body mass index. Dependent variable: nighttime diuresis rate (mL/min). Data are expressed as standardized regression coefficients (β) and * *p* values < 0.05.

**Table 3 jcm-09-02532-t003:** Commonly eaten protein-rich foods with their sodium content and their estimated effect on urinary osmolality by urea and sodium, per 100 g of edible portion.

Food	Protein (gram) ^1^	Sodium (milligram) ^1^	Effect on Urinary Osmolality (mOsm) ^2^
**Meat products**			
Bacon, smoked, salted	22.7	1532	263
Beef, smoked, fillet	22.9	3100	265
Chicken ham	20.0	787	148
Salami	19.8	1370	173
Sausage, poultry, raw	18.9	310	121
Spread, meat salad	8.0	761	79
**Milk products**			
Cheese spread, full fat	11.0	702	92
Cheese, Cheddar	25.5	700	173
Cheese, Cottage	11.7	370	81
Cheese, Emmentaler	28.0	300	170
Cheese, Mozzarella	19.5	200	118
Cheese processed, slices light	17.5	1253	152
**Cereal products**			
Baguette, white	8.5	486	74
Bread, French roll	12.5	518	101
Bread, white	8.4	484	74
Croissant	9.2	426	76
Crackers	11.3	795	105
Corn flakes, enriched	8.0	613	77
Muesli, crunchy, enriched	7.5	320	61
Granola, crunchy, original	8.6	98	58

^1^ Protein and sodium content of the foods were from the Belgian food composition database (Internubel.be). ^2^ Calculation of effect of urine osmolality: urea osmotic load (mOsm) + sodium osmotic load (mOsm) = total osmotic load (mOsm). urea osmotic load: ([Protein (g)/6.25] × 2.14)/60 × 1000; conversion coefficient from nitrogen to protein for meat products 6.25, milk products 6.38, cereal products 5.70 [17]. sodium osmotic load: sodium (mg)/23 × 2; as each sodium ion has its negative counter-ion contributing to the osmotic pressure of urine.

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
