# Peer review of "Could Evening Dietary Protein Intake Play a Role in Nocturnal Polyuria?"

_jcm, 2020, doi:10.3390/jcm9082532_

Round 1

Reviewer 1 Report

The authors compare urine composition in q3h samples obtained from patients with and without nocturnal polyuria (defined as  nocturnal urine volume>1/3 urine total urine volume).  The paper is well written and the methods are clearly described.  

I have the following suggestions for improvement of the manuscript:

1) Data  presented in figure 1 omits half of the data for NP subjects (it appears like the figure was incorrectly cropped)

2) Figures 2,3 and 4 cut off 2 data points during the night -- again incorrectly cropped?

3) Figure 2 provides data on urine volume, free water clearance, and urine osmolality.  A fourth graph illustrating osmolar clearance should be included

4) The authors state in the methods that urine volumes were validated by determining urine creatinine excretion.  However, no data are provided.  Were there any differences between creatinine excretion rates in the 8 collection periods?  The reader needs to be convinced that the higher urine volume at night in the NP group is not due to inaccuracies in urine collection. 

5) The discussion acknowledges that dietary protein intake was not determined but it still tends to equate urine urea excretion with protein intake.  I believe that the conclusions outstrip the data.  Specifically, there should be discussion of the phenomenon of urea exaltation (see for example: GOLDSTEIN, MARVIN H., PAUL K. LENZ, AND MARVIN F.LEVITT. Effect of urine flow rate on urea reabsorption in man: urea as a “tubular marker. ” J. Appl. Physiol. 26(5): 594-599. 1969.).  During a water diuresis, there is an brief increase in urea excretion which is explained by decreased urea reabsorption in the papillary collection duct and loss of urea from the renal medulla.  Thus the authors findings could be explained by a decrease in vasopressin levels (or action) at night in the NP group, with increased urea excretion being the result, rather than the cause of increased urine output.  This could be tested by administrating DDAVP to both groups to see if the higher levels of urea excretion in the NP group persist.  Dietary history is also needed to determine the cause of the authors' findings. 

Reviewer 2 Report

This manuscript entertains the idea that protein-rich meals in the evening may contribute to nocturnal polyuria through an increase in urine osmolality by increased nocturnal urea excretion. The study is based on the analysis of 24h urine samples from 52 control subjects and 118 patients with nocturnal polyuria. Patients with nocturnal polyuria had greater nocturnal sodium and urea excretion than control subjects, suggesting that a greater nocturnal urea excretion contributes to nocturnal polyuria.

General Comments

This study addresses an interesting and potentially clinically relevant hypothesis. The idea that protein-rich evening meals may contribute to nocturnal polyuria is certainly intriguing.

Specific Comments

1. The lack of food intake data is a major problem that limits the conclusions that can be drawn from this study. I don’t think the conclusion that protein-rich evening meals contribute to nocturnal polyuria can be made without food intake data. However, the data seem to indicate that nocturnal urea excretion is higher in patients with nocturnal polyuria and, therefore, patients with this condition may be advised to avoid protein-rich evening meals. However, the authors need to acknowledge that the available data form their study does not allow to conclude that the higher nocturnal urea excretion in the patient group compared to the control group is due to greater evening protein-rich meals.

2. Nocturnal osmolality was less in the patient group compared to the control group (Table 1). How does this finding fit into the hypothesis that a greater nocturnal urea excretion contributes to nocturnal osmotic diuresis?

3.  Since the authors hypothesize that greater nocturnal urea excretion contributes to nocturnal polyuria via an increased osmolality, it may be more appropriate to present urea concentrations (mmol/mL) rather than total urea excretion (in mmol/h). Since the greater nocturnal urea excretion (in mmol, Table 2) was associated with greater nocturnal urine production, the greater urea excretion may not result in an increase in osmolality and, thus, may not contribute to osmotic diuresis in patients with nocturnal polyuria. I strongly suggest to include a figure showing the urine urea concentrations for all 8 urine samples.

4. The figures need to be improved to show the x-axes and the data from all 8 urine samples.

5. Where patients with signs of congestive heart failure excluded?

6. At which time point was the blood sample taken? How valid is the time course of the free water clearance (Fig. 2) if only one blood sample was used for these calculations? Ideally, a blood sample would be needed for each time point.

Round 2

Reviewer 2 Report

The authors have appropriately addressed all my previous comments. The manuscript would benefit from proof-reading by a native English speaker.